# Urban Air Pollution, Urban Heat Island and Human Health: A Review of the Literature

Awais Piracha [1,*] and Muhammad Tariq Chaudhary [2]

1 School of Social Sciences, Western Sydney University, Penrith, NSW 2751, Australia
2 Department of Civil Engineering, Kuwait University, Shadadiya 13060, Kuwait; tariq.chaudhary@ku.edu.kw
* Correspondence: a.piracha@westernsydney.edu.au

**Abstract:** Many cities of the world suffer from air pollution because of poor planning and design and heavy traffic in rapidly expanding urban environments. These conditions are exacerbated due to the Urban Heat Island (UHI) effect. While there have been studies linking the built environment and air pollution with health, they have ignored the aggravating role of UHI. The past urban planning literature in this field has also ignored the science of materials, vehicles and air pollution, and technological solutions for reducing cumulative health impacts of air pollution and UHI. Air Pollution, built environment and human health are complex discussion factors that involve several different fields. The built environment is linked with human health through opportunities of physical activity and air quality. Recent planning literature focuses on creating compact and walkable urban areas dotted with green infrastructure to promote physical activity and to reduce vehicle emission-related air pollution. Reduced car use leading to reduced air pollution and UHI is implied in the literature. The literature from technology fields speaks to the issue of air pollution directly. Zero emission cars, green infrastructure and building materials that absorb air pollutants and reduce UHI fall within this category. This paper identifies main themes in the two streams of urban air pollution and UHI that impact human health and presents a systematic review of the academic papers, policy documents, reports and features in print media published in the last 10–20 years.

**Keywords:** urban transport; air pollution; urban heat island; human health; built environment

## 1. Introduction

### 1.1. Air Pollution and Urban Heat Island Effect in Contemporary Cities

During the COVID-19 lockdown in 2020, the snow-covered peaks of the Himalayas were visible from Punjab in India for the first time in 30 years. It was a joyous occasion for people in the region who could enjoy much better air quality. Indian citizens were not alone in being able to breathe easier: the experience was shared in many other parts of the world [1]. Air filled with smog and pollutants in numerous cities harms the health of millions of their residents [2].

Humanity's migration from rural to urban areas has been a continuous phenomenon that accelerated with the advent of industrial age in the 18th century. The tipping point between the urban–rural demographic balance was reached in 2008 when the global urban population outnumbered the rural one and is projected to be more than 70% of the global population by 2050 [3]. Some salient features of urbanization are intensive infrastructure setups, built-up areas, paved surfaces, transportation networks, traffic and human congestion, significant temperature difference between urban and surrounding rural areas and the absence of natural habitat and green open space. It is worth noting that vector borne and infectious diseases (i.e., cholera, dysentery, TB, and typhoid etc.) common in early industrial cities of the 1800s are now replaced with chronic diseases (cardiac, pulmonary, psychological and cancer) in more affluent cities, which is mainly attributed to sedentary lifestyles and exposure to air pollution.

Although urbanization has significantly improved the quality of life and human comfort, it has also resulted in some unintended problems. Air pollution and urban heat island (UHI) are two such phenomena. Urban air pollution from mass car use is relatively recent. It is a by-product of the normalization of suburban living or sprawl. The last century has witnessed dramatic changes in transport technologies where compact cities of the past have expanded outward to low-density car-dependent suburbs. The era of ubiquitous motor cars can be blamed for many of the air pollution troubles of contemporary cities. UHI is peculiar to large cities by which the temperature in the urban center is noted to be higher than the surrounding rural or natural areas for all major metropolis except for large desert cities. Urban heat, labeled as an invisible hazard, has the highest potential for human fatalities in some countries due to the climate change effects [4]. During the 1900–2021 period, heat waves around the globe resulted in 171,856 fatalities [5]. Heating up of the urban environment is recognized as a serious health threat in many cities of the world.

### 1.2. Urban Planning and Human Health

Urban planning plays a crucial role in designing urban facilities and features that impact health and wellbeing of the local communities. New design perspectives of Smart Growth, New Urbanism and Transit-Oriented Development aim to create convivial and accessible neighbourhoods that support non-motorized transportation and connect other parts of the city with transit [6]. Influential writers such as Jan Gehl have for decades advocated cities designed for people and that are free of cars [7,8]. Peter Newman advocated higher densities and a transport system reliant on trams, metro lines and heavy rail [9,10]. These are visions of cities full of people meeting each other and enjoying life, with wide footpaths and cycle lanes that are safe from cars and reliant on an accessible public transport system. These is also a vision of cities free of air pollution from cars and plentiful opportunities for exercise.

Unfortunately, many fast-growing cities suffer from heavy traffic, noise, air pollution and unsafe roads for non-motorized transport. These conditions have severe adverse effects on the health of their citizens. Topics such as air pollution, the built environment and human health are involved in complex debates that touches on the disparate fields of urban planning and design, environmental engineering, public health, transportation planning and engineering and trees and plants ecology. Elements of these and other fields form the pieces of a jigsaw puzzle that constitute a healthy city free of air pollution. Air pollution is complex. It is a concoction of various gases and particulate matter. Air pollution includes particulate matter, volatile organic compounds, ozone, carbon monoxide sulphur oxide and nitrogen oxide [11]. UHI exacerbates the impacts linked with air pollution [12]. The documented adverse effects of UHI are related to higher energy and water consumption [13–15], deteriorated air quality [16,17], higher $CO_2$ emissions [18] and serious implications for human health.

Human health is linked with the built environment and air pollution through opportunities of physical activities for good health and air pollution inhalation leading to poor health. The focus of the planning literature is on improved planning and design to reduce car use and create more opportunities for physical activity. A reduction in air pollutants is only implicit in the literature. Another body of literature from science and technology fields addresses the issue of air pollution more directly. Improvements in fuel and in car emissions standards, electric cars and growing vegetation that absorbs air pollutions or serve as buffers are examples of the literature. This second body of literature is particularly relevant for existing car-oriented neighborhoods. Because of the historic legacy, most suburban areas in many cities will remain car dependent. Moreover, the desire for cars is unlikely to end anytime soon. On the other hand, urban planning and public policy efforts to address the impact of UHI are relatively recent. However, rapid progress is being made to address the threats caused by UHI through legislation, policy papers and the adoption of heat-beating measures of reflective roofs, cool pavements, urban forests, blue features and wind alleys, etc. Urban planners and public officials are teaming up with research communities for finding sustainable solutions to the emerging nexus of urban air pollution and UHI effect and its implications on human health [19].

### 1.3. Organization of the Paper

The paper identifies key themes within the above-mentioned two streams and provides a review of academic papers, policy documents, various standards and features in print and electronic media published mostly in the last 20 years. In reviewing that material, emphasis is on the contributions of key researchers that have greatly influenced their respective fields of study related to this research. In addition to the relevant planning and design literature, the paper consults ecological, air pollution science, UHI and vehicle science-related material as well. A total of 171 documents were reviewed out of which 38 and 72 documents were related to the problems caused by urban pollution and UHI and their impact on human health, respectively. Urban planning, public policy and other measures to address urban air pollution and UHI impacts were discussed in 41 and 27 documents, respectively. Documents related to both urban air pollution and UHI were tallied in both categories. About 75% of the reviewed documents were less than a decade (since 2012) old. Only 7% of the documents were from 2000 and earlier. Figure 1 presents a graphical summary of the reviewed documents categorized according to the four theme topics and the time period of publication.

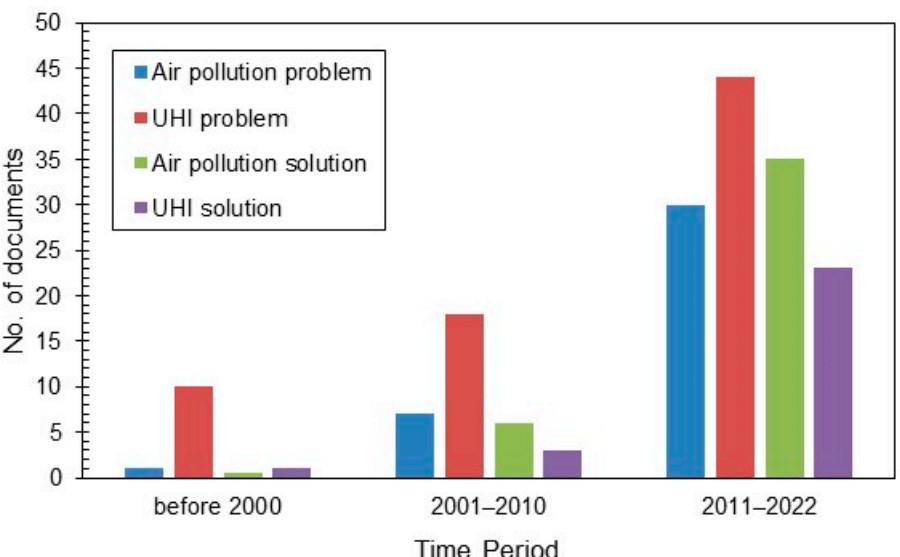

**Figure 1.** Time period of documents reviewed in the study.

Following the introduction, the paper explores the extent of motor vehicle penetration, the nature of air pollution and UHI and how the two phenomena harm human health. It then presents a review of the built environment literature as it relates to air pollution, UHI and human health. In the section that follows, a review of material on planning, public policy and technological solutions to reduce impact of air pollution and UHI exposure to human health is critically examined. Lastly, a concluding section sums up the findings and points to research gaps in the field.

## 2. Understanding the Health Problems Caused by Urbanization

### 2.1. Transportation in an Urbanized World

The total number of vehicles in the world was estimated at 1.45 billion in 2022 [20]. This number grew from 670 million in 1996 and from just 342 million in 1976 [21]. At that rate of growth, the world will have about 2.8 billion vehicles in 2036. In almost all countries of the world, the vehicle numbers and total kms travelled by vehicles have been growing. This trend is much more pronounced in fast-growing economies such as China. For a long period of time, the United States had the most cars. She had a vehicle fleet of 276 million and annual car sales of about 17 million in 2019 [22]. However, in recent years, China has overtaken the USA as the country with the most cars. In 2017, it had more than 300 million cars and annual cars sales of more than 27 million [23].

Globally, there were 10 million electric cars on the roads in 2020 [24]. The electric car sales grew by 41% in 2020 despite the COVID-19 pandemic. With existing EV policies, the worldwide EV cars fleet is forecasted to grow to 145 million by 2030 [24]. Electric cars do not discharge any pollutants harmful to human health in cities. Their sources of electricity can be polluting. However, even the non-renewable sources of electricity, such as gas and coal, are located away from dense cities and can reduce pollution easier. It is easier to control pollution at a few powerplants than for millions of cars. Moreover, the electricity production with renewable resources such as solar and wind is growing phenomenally. IEA (International Energy Agency) estimates that, by 2026, almost 95% of the increase in global power capacity will be from renewable sources [25].

### 2.2. Urban Transport and Air Pollution

Air pollution in cities is a well-known problem. The sources of air pollution in cities include heating, cooking, industrial processes, power generation and transport. The type of pollution experienced by urban areas vary from the developing countries to the developed countries. In developing countries, the main sources of air pollution are cooking and industries. In developed countries, cars are a much bigger culprit [26].

Anenberg et al. [11] report that 84% of transportation emission-related deaths took place in the G20 countries. Only four of the largest vehicles markets in the world, i.e., China, India, the European Union (EU) and the United States, were responsible for 70% of these deaths. Such deaths have been declining in the EU and the USA. However, they are increasing in China and India. The death toll from transport-related pollution is also increasing in most other parts of the world [11].

Global transportation-related emissions are fueled by economic development resulting in higher vehicle ownership and increased industrial and freight activity while the counter measures for reducing air pollution include better engine technologies and stricter car and fuel emissions standards [11]. The best overall indicator of a city's health is its air quality [27]. Air pollution levels are low in cities designed well and that have good public transport, streets for pedestrians and green infrastructure to filter and buffer pedestrians from air pollutants. Neira [27] reports that air quality in 80% of cities in the world is worse than the World Health Organization's (WHO) air-quality limits.

The emissions from motor vehicles are made up of a toxic mix of carbon monoxide (CO), nitrogen oxides (NOx), sulphur oxides (SOx), particulate matter (PM10 and PM2.5), many volatile organic compounds (VOCs) and ozone. USEPA [28] report presence of up to 1162 different compounds in vehicles' emissions. The emissions from vehicles have adverse impacts on the health of millions of people, particularly those living close to busy roads. The vehicular emissions are also responsible for the acidification of lakes and streams [29].

Vehicular transport not only impacts air quality directly through emissions but also indirectly as a vital agent of climate change and a contributor to UHI. Transport is responsible for about one-fifth of global greenhouse gas (GHG) emissions [30]. Ebi and McGregor [31] argue that climate change also affects air pollution through modifications in temperature, rainfall and wind. These meteorological factors impact air pollutants in their formation, chemical changes, travel and spread.

Kinney [32] reviewed the literature that examines how climate change impacts air quality and human health. He discovered that the two most harmful air pollutants of the transport origin, i.e., ozone and PM2.5 (particulate matter smaller than 2.5 μm), have higher concentrations at higher temperatures. He reports that several thousand additional deaths from ozone exposure are expected in the United States because of climate change. Wild (bush) fire frequencies and intensities are increasing due to climate change. Bush fires are a major source of small particulate matter and are, thus, a significant health risk [32].

### 2.3. Air Pollution and Human Health

Transportation attributable air pollution resulted in 3.5 million premature deaths in 2017 induced by respiratory infections, diabetes, cardiovascular disease, lungs infection,

obstruction and cancer [11]. In addition to exhaust-pipe emissions, vehicles also contribute to air pollution through petrol evaporation, resuspending dust and particles from the wear and tear of rubber tires. Anenberg et al. [11] also report that transportation emissions caused about 8 million years of life lost and USD 1 trillion equivalent health damages worldwide in 2015. Most of the loss of life is attributable to ozone and PM2.5 particulate matter. Burning fossil fuels are reported to cause 3000 premature deaths in Australia alone every year [33].

USEPA [34] explains that ozone ($O_3$), a very strong oxidizing agent, inflames and harms the respiratory system by constricting the muscles in the airways, leading to asthma, bronchitis and infections in healthy people and worsening the conditions of people with lung diseases. Small particulate matter PM2.5 penetrates deep into the lung and even passes through various membranes to join the bloodstream. PM2.5 can cause asthma, bronchitis and heart diseases [35].

Air pollution resulting from high volumes of motor vehicles in cities causes respiratory health effects in the community. A Swiss research study discovered that air pollution from motor vehicles increases attacks of wheezing with breathing, breathlessness and frequent coughing [36]. Yujing et al. [37] argued that alternative modes of transport such as electric vehicles are needed to improve the public health of communities.

In addition to air pollution, cars hamper human health through noise and injuries resulting from motor vehicle road accidents. Road traffic accidents are the eighth leading cause of death globally. Traffic accidents kill 1.35 million people and cause 50 million injuries each year [38]. Many of them are in urban areas. Injury in traffic accidents is the number one cause of death for 2–29-year-olds. The traffic injuries and deaths are disproportionately higher in vulnerable road users and residents of poorer countries. The death rate from traffic accidents is three-times higher in low-income countries compared to high-income countries [38].

Greenhouse Gas (GHG) emissions from cars and resulting climate change is another pathway through which transportation-related air pollution harms human health. Dean and Green [39] reviewed academic material on air-quality-related health impacts caused by climate change in Sydney. They also looked at the related research gaps. There is a significant overlap between sources of climate change and sources of air pollution. Both are often related to the combustion of fossil fuel. Efforts to reduce sources contributing to climate change will reduce air pollution as a side effect. Beggs and Bennett [40] found that instances of asthma and allergies will increase due to climate change. However, Dean and Green [39] report on the absence of studies that look at air-pollution-related human health problems in future climatic conditions in Sydney.

Ebi et al. [41] carried out a comprehensive review of the literature on health risks related to climate change. They report their findings on several pathways of health impacts. First and foremost, additional heat related to climate change is linked with a very high confidence in an increase in stress, injuries and death. Climate change is also a source of the reduced availability of water and decreased food production. This would have a significant health impact on many communities, especially the less effluent ones. Climate change is also likely to increase the instance of vector and food borne disease. Diseases and mortality caused by colds, on the other hand, are likely to decrease moderately [41].

While the negative impacts of air pollution on physical health are widely reported, the impacts of the same factors on mental health are under-researched. However, some epidemiological studies indicate links between air pollution and poor mental health. Air pollutants are associated with anxiety, depression, poor cognitive development in children, dementia and psychosis [42]. Several studies [43–45] point out that the neuroinflammatory responses caused by the air pollutants are responsible for a decline in mental health.

### 2.4. Built Environment and UHI

UHI was first systematically observed in the early 19th century in London [46]. Since then, UHI has been extensively studied through climatological [47], anthropogenic [48],

urban planning [49–51] and engineering [52–54] lenses to understand its physical processes, contributing factors and the effectiveness of mitigation measures.

Tzavali et al. [55], Kershaw [56], Stewart and Mills [57], Rizwan et al. [58], among others, conducted excellent reviews on the phenomenon of UHI. Extensive studies on UHI intensity are available for the majority of the populous urban centers of the world by utilizing land-based as well as remotely sensed thermal data e.g., [59–66]. These studies reveal that, except for cities located in deserts (e.g., Phoenix, Jeddah, Riyadh, Kuwait, Abu Dhabi, Mousal, etc.), UHI is mostly positive for large cities. UHI intensity generally aligns with an increase in a city's latitude as well as population and development level [67–70]. UHI intensity exhibits diurnal as well as seasonal variations with maximum recorded intensities of close to 10 °C and an average value of 2–4 °C [60].

According to past studies, the two main sources of generating heat in an urban area are solar radiation and anthropogenic heat. Solar radiation warms the atmosphere as well as the urban surfaces through direct solar heat (DSH). Roofs and pavements constitute about 60–70% of surface area in a typical city in a developed economy [71]. These surfaces are usually dark with low solar reflectance (or albedo) and a part of DSH is stored inside these urban structures and is released into the atmosphere as indirect solar heat (ISH) when the contribution of DSH diminishes. On the other hand, anthropogenic heat is produced by agents of human activity such as vehicle exhaust, appliances, building operations (heating, cooling, lighting, etc.), transportation activities, power plants, etc. As DSH is constant for a given urban or rural area, the main contributor to UHI, therefore, seems to be ISH and anthropogenic heat. ISH is responsible for nighttime UHI in desert cities, which exhibit the 'urban cool island' effect during the daytime [72]. In the absence of an efficient public transportation system and heavy reliance on cars, vehicle exhaust can be a significant contributor to the anthropogenic heat of a city [73,74].

*2.5. Human Health Impacts of UHI*

Health impacts of high temperatures are well documented in the medical literature, e.g., Shattuck and Hilferty [75], Gover [76] and Basu [77]. With increasing global urban population, the exposure to heat-related health risks in metropolitan cities and larger urban centers has proportionately increased [78]. A recent study, based on data from 1300 cities across the globe, estimates that close to a quarter of the world population (i.e., nearly 1.7 billion persons) is exposed to extreme heat [79]. It is also noted that human health implications of heatwaves are relatively severe in mild and cold climates as compared to the warm ones [80]. Barrow and Clark [81] noted that mortalities attributed to heat stress are under-reported because heat stress is the driver for apparent causes of death due to cardiovascular, respiratory and cerebrovascular failures. Additionally, high temperatures are also attributed to increased mental health emergencies [82]. Vaidyanathan et al. [83] and De'Donato et al. [84] attribute heat stress as the leading reason for weather-related mortality in the USA and Europe. Macintyre and Heaviside [85] estimate the contribution of UHI to heat-related mortalities in a European city to be as high as 40% during heatwaves.

Human body reacts to elevated temperatures through two main processes [86–88]. First is an increase in blood supply to the skin in order to dissipate the excess heat and the second is the production of sweat for cooling the skin. The redistribution of blood to the skin leads to an increased cardiac demand for oxygen requiring the heart to pump harder. The cardiovascular strain produced to meet the oxygen demand may lead to cardiovascular collapse that may result in death [89]. Similarly, heat-related hyperventilation increases pulmonary stress, resulting in lung injury or failure. There is also a risk of heat stroke and other cerebrovascular impairments due to prolonged body temperature in an excess of 41 °C. Dehydration caused by the heat stress can also result in kidney damage or failure. The severity of these episodes is usually increased for vulnerable population groups, i.e., children, elderly and people with pre-existing conditions [90].

Henschel et al. [91] observed that the rate of heat-related mortalities was significantly higher (more than five times) in larger cities compared to the surrounding rural areas.

Clarke [92] was perhaps one of the earliest to relate such mortalities to the microclimatic changes brought by the level of urbanization. He attributed the higher mortality rate during urban heatwaves to the higher difference in the nocturnal urban–rural temperature difference that was noted to be in the range of 2–4 °C for two US metropolitan cities, i.e., New York City and St. Louis, MO, during the heatwaves of 1955 and 1966, respectively. The analysis of the excess mortalities during these heat waves attributed 30 and 5 deaths per 100,000 to heat-related ailments in the cities and the rural areas, respectively. A critical analysis of the health impacts of heatwaves in Europe [93–95], USA [96,97], Japan [98], Australia [99] and China [100,101] strongly correlates the level of urbanization to the numbers of excess fatalities during these climatological events. This contrast highlights the effects of the level of urbanization and UHI on heat-related morbidity and mortality on a global scale.

Urban features that exacerbate the level of morbidity/mortality during extreme heat events above and over the biophysical factors include some key characteristics that are attributed to older housing stocks such as congestion, a lack of ventilation and an absence of cooling [102]. Such housing units are also more likely to be located in areas possessing dated urban planning aspects such as a proximity to sources of anthropogenic heat from industry and traffic, the sparseness of vegetative cover and being located away from a green or blue area [103].

The combined impact of urban pollution and urban heat on human health has also been explored. Lai and Cheng [104] report that, in the warm center of metropolitan areas, the UHI in conjunction with transport-related air pollutants increases hospital respiratory admissions. Grigorieva and Lukyanets [105] reviewed data from multiple countries and found a compelling link between air pollution, atmospheric temperature and respiratory illness. Perera and Nadeau [106] found a similar nexus for children health in USA. Shirinde and Wichmann [107] observed an increase in mortality rate due to respiratory ailments caused by air pollution with increased temperature in South Africa. Sabrin et al. [108] developed a human health vulnerability index that included the UHI effect and $O_3$-PM2.5 pollution for a neighborhood in New Jersey. The index indicated the highest vulnerability in socially deprived areas and recommended appropriate interventions suggested by city planners.

## 3. Urban Planning Concepts for Improving Human Health

Concepts that advocated higher densities and compact development to make cities more livable and walkable, such as TOD, Smart Growth, New Urbanism, Healthy Cities and so forth, have now become commonplace in the planning literature. There are obvious health benefits in walkability. In addition, it is implied in these concepts that less reliance on automobiles will reduce air pollution [109]. However, how do these spaces fare in terms of human health?

Higher densities in cities make public transport more viable. Peter Newman and his colleagues link compact cities with more public transport use with improved human health [110]. It is known from Rissel et al. [111] that people who use public transport carry out about 15 min more exercise than those who rely on their cars for their transport. Newman and Kenworthy [9,10] argue that large volumes of the literature are available on human health impacts of urban forms and transports. Jan Gehl [7,8] advocates that cities are for people and not for cars. He designs open spaces that are inviting and attractive places for people to hang out and meet each other. Reduced air pollution and more physical activity and, hence, better health are implicit in his recommendations.

Stevenson et al. [112] used a health impact assessment framework to estimate the population health effects of a compact mixed-use city with good access to public transport and a modal shift away from cars to non-motorised transport for eight large international cities. Their model indicated that a compact city would result in a reduction in cardiovascular and respiratory diseases and diabetes for all cities. They predicted overall health gains of 420–826 life-years per 100,000 population for compact cities.

There is a large body of the literature that focuses on healthy cities/healthy built environments. The first and the foremost is the WHO Healthy Cities program. It is

a broad-based global drive that works with governments of hundreds of cities across the globe, helping them place high emphasis on health in all aspects of their city governance: social, economic, environmental and political [113]. There are numerous public and not-for-profit sector reports and programmes that examine the link between urban planning/neighbourhood design and health mostly through the provision of transit and increased physical activity. Extensive academic research has been published in this area that is typified by [114].

Smart Growth, a US EPA program, is about building compact and walkable, self-sustaining and attractive neighbourhoods. It rests on the principles of mixing land uses, compact design, walkability, preserving open spaces and providing transport choices [115]. Smart Growth is routinely adopted as a measure in city planning to reduce air pollution [116,117]. However, Gren et al. [118] argue there is not enough evidence is available to confirm air pollution reduction by Smart Growth.

New Urbanism resembles Smart Growth but is a more architecturally oriented approach to urban development. It recommends walkability, mixed land use and accessible public spaces. Moreover, similarly to other similar approaches, New Urbanism focuses on the human scale [119]. New Urbanism recommends gridiron street pattern, narrow streets, footpaths on both sides of the streets, smaller lots, shallow setbacks, and porches instead of driveway and garages [120]. New Urbanism has become a very influential concept and has been applied to the development of urban areas across the world. Iravani et al. [121] contended that New Urbanism results in reduced car use and, thus, results in less exposure to noise and ambient air pollution. Wu et al. [122] report that the literature on the link between new urbanization patterns in China and ambient air pollution is inconclusive. The new urbanization scheme in China has underlying principles similar to New Urbanism. Wu et al. [122] examined environmental data from 113 major Chinese cities for the period 2013–2017 to explore the link between new urbanization and air quality. They discovered some associations between new urbanization and improvements in air quality. However, this link was not very strong.

Van der Waals [123] discovered something similar. He reviewed the environmental performance of compact cities in the Netherlands. He discovered that the actual environmental benefits of New Urbanist-type urban developments are higher at the conceptual levels than in their actual performance. Lund [124] argues that the environmental performance of New Urbanist developments depends on the behaviour of people residing in those schemes. Air pollution in such areas will only be less if people reduce their car use.

Recently, Iravani et al. [121] reviewed the literature on health impacts of New Urbanism. They found that compact and mixed-use New Urbanist planning resulted in positive health impacts for all demographics in American cities. Ewing et al. [125] discovered that compactness measures in cities result in reduced BMI, obesity, heart disease, high blood pressure and diabetes. The reduction results from functional physical activity in the form of active travel to work, shopping and other destinations.

Transit-Oriented Development (TOD) is another related concept that prescribes a provision of transit nodes in city centers or new developments. Higher densities and mixed-use are recommended for those nodes. These nodes are then connected to one another by transit buses, light rail or metro lines. Such arrangements increase walkability and reduce car use and, hence, are beneficial for communities' health. Noland et al. [126] argue that TOD, due to reduced automotive use, results in a reduction in air pollution in cities. While this intuitive link is often mentioned in conceptual studies, more exploratory research is needed for establishing a clear link between TOD and air pollution outcomes.

Zhou et al. [127] investigated the link between urban form and air pollution for the fast-growing and large Chinese cities of Beijing, Tianjin, Shanghai, Chongqing and Guangzhou. For their research, they examined data from 2000 to 2012. They discovered that urban expansion is associated with a reduction in air pollution. They also found that fragmented cities experience less air pollution.

Liang and Gong [128] examined the effects of urban form on air quality in cities of different sizes and development levels in China. They discovered that cities with spatially connected urban areas experience more particulate air pollution. Similar observations were noted by Carozzi and Roth [129] for urban areas in the USA while investigating the link between urban density and city residents' exposure to ambient air pollution. Castells-Quintana et al. [130], however, in their global study of density and air pollution discovered that higher density cities are associated with lower particulate air pollution per capita.

It is clear from the discussion in this section that the theoretical benefit of reduced air pollution from compact city prescriptions is often not fully realized in practice. In some cases, higher densities are associated with even higher levels of air pollution. It seems that the behavioral aspects of the residents of the compact cities are also a very important factor in determining the level of air pollution [123]. It is also obvious from the above that not enough research has been conducted to measure actual reduction in air pollution by applying Smart Growth, New Urbanism, TOD and other compact urban-design perspectives.

## 4. Technological Solutions to Reduce Air Pollution

Technological solutions for reducing air pollution emitted by cars are highly relevant for improving air quality in cities. In the following, a discussion is presented on vehicles that do not emit any air pollution, the Green Infrastructure (GI) that can absorb air pollution and/or serve as barriers and building materials that can absorb air pollution.

### 4.1. Electric Vehicles and Hydrogen Cell Vehicles

Electric and hydrogen-cell are zero local emissions (at the tailpipe) vehicle technologies. However, they need different and more ubiquitous fueling infrastructure in the urban landscape. These cars can not only improve air quality in newly designed compact cities but, more importantly, also reduce air pollution in the large swathes of low-density suburban areas in many cities of the world. Given their high level of relevance and the need for providing opportunities for recharge in the urban landscape, a better understanding of these technologies is essential for urban planners and designers.

In addition to obvious human health benefits in cities, the environmental benefits of electric vehicles (EVs) include higher energy efficiency, reduced dependency on fossil fuels and reduced noise [131]. The rapid adoption of electric vehicles in numerous countries of the world for their health and climate change benefits is perhaps going to be the most profound impact on the future of air pollution (reduction) in the cities [24].

Bradley and Frank [132] elaborate on the environmental sustainability credentials of electric cars. Electric vehicles do not use fossil fuels and, thus, significantly reduce carbon emissions. They also have longer lives because of the durability of electric motors. The battery technology of these cars is constantly improving to cover longer distances before recharge, and they are becoming longer lasting and cheaper. Electric vehicles are a better choice when it comes to addressing climate change causing Greenhouse Gases (GHGs) when they are recharged with renewable electricity sources such as solar, wind or hydro power [133].

Soret et al. [134] studied air quality reductions in electric vehicles in Barcelona and Madrid. They discovered that electric vehicles use results in improved air quality through a significant reduction in nitrogen oxides and carbon monoxide. Malmgren [135] estimated the societal benefits of electric cars. She identified and assessed seven different social benefits of EV in monetary terms. She estimates the overall social benefit to be about CAD 16,000 over a 10-year 120,000 miles life of an electric car. The author attributes one-tenth of that benefit to health improvements. Despite the obvious air pollution reduction and general sustainability credentials, hesitancy in electric car adoption remains due the range anxiety and a lack of proper recharging infrastructure [24]. Urban planners and designers can play a strong role in providing opportunities and locations of car charging stations in their plans and designs for urban development to reduce the range anxiety.

Electric vehicles can be a core element for urban planners and designers in creating ecologically sustainable urban neighbourhoods. A Community Mobility Hub was deployed in Austin, Texas, in 2018 that, in addition to walkability, bike paths and mixed use, includes a charging hub for shared electric bikes and scooters and EV-charging car parking spaces [136]. Similar initiatives are springing up elsewhere in the world.

Hydrogen fuel cells is another technology that does not discharge any pollution from cars' tailpipes. Hydrogen is an excellent renewable energy source. Similarly to electricity, it is a very effective energy carrier [137]. Hydrogen Fuel Cell Electric Vehicles (HCEVs) use a technology that converts hydrogen into electricity. As a result, the HCEVs' discharge only water vapor and warm air [138]. In HCFVs, the fuel cells produce direct currents (DCs) to run the electric motor that moves cars. The hydrogen fuel cell is typically paired with batteries and regenerative brakes that can produce electricity. The cars need only a small fuel tank and can be refilled with hydrogen quickly. They have a long range once filled with fuel [138]. HCEVs are predicted to be a major part of the future car fleet–technology mix [137]. Their current availability and market penetration is very low due to a lack of hydrogen fuel refilling infrastructure [138]. Urban planners would have to allocate provisions in their plans and designs for the refilling infrastructure.

*4.2. Urban Green Infrastructure*

Plants can be used to both absorb air pollution and to serve as barriers between people and car related air pollution. Prashant Kumar has written widely on air pollution dispersion, its suppression and its impacts on human health. Barwise and Kumar [139] explored the literature on how green infrastructure such as vegetation barriers can be used to improve air quality around roads with large volumes of traffic. They evaluated the design features of barriers and species types that are the most effective in reducing air pollution from traffic. They discovered that different biological traits of plants are effective in reducing different pollutant types. Overall, the removal of transport-related air pollution is a function of small leaf size, high leaf complexity and optimal vegetation height and density. In general terms, taller vegetative barriers (between traffic and people) are more useful in case of open roads and shorter for street canyons. Equipped with that knowledge, they have formulated a plant-selection framework for reduced air-pollution exposure.

Abhijith and Kumar et al. [140] reviewed the literature on improvements in air quality by green modifications in open roads and built-up street canyon environments and came up with recommendations for green infrastructure design. They discovered that, in the case of street canyons, low-level green plantings such as hedges helped in improving air quality. In the case of open roads, wide and tall vegetation was more effective in reducing air pollution. Moreover, for reducing ultra-fine particles (smaller than 0.1 μm), tree species with strong trichomes (sticky glades on flowers), ridges and groves are more effective. For reducing sub-micrometre particles (larger than 0.1 μm) and gaseous pollutants, stomata on leaves (mechanism responsible for photosynthesis) are more relevant.

Maher et al. [141] conducted a trial by planting silver birch trees outside a row of terraced houses adjacent to a main road in Lancaster, UK. The objective was to reduce small particles from the air that can penetrate deep into lungs as well as enter bloodstreams and harm other organs. They discovered that the planting of hedges and silver birch trees as barriers at a busy road reduced particulate pollution by more than fifty percent. The particulate matter captured on leaf hairs and within the leaf was observed on both the road-facing and non-road-facing sides of the trees. Maher et al. [141] argue that the benefit of roadside trees in reducing particulate matter is underestimated in the literature.

Jeanjean et al. [142], however, discovered that the contribution of plants in reducing air pollution is limited. They modelled the air quality for a section of London and discovered that tree leaves reduced small particles (PM2.5) by only 1–2 percent. A larger impact of trees was, however, noted as barriers on the dispersion of pollution. As noted earlier, various heights and configurations of green infrastructure contribute as barriers for different street conditions.

Zhang et al. [143] explored the cleaning effects of rainfall on leaves and branches of four tree species and one bush in Beijing, China. They wanted to study if leaves can be cleaned by rainfall and reused for filtering air pollution. They discovered a wide difference in particulate matter removal among different species due to leaf properties. The London plane tree (*Platanus acerifolia*) had the highest particulate matter removal because of its rough leaves. Predictably, Zhang et al. [143] also found that high-intensity rain was needed to clean the leaves of such plants.

*4.3. Building Design and Materials*

Building designs and materials can also play an important role in reducing air pollution. Intense research is ongoing in this field. European Commission (EC) [144] provides an account of an EU-funded research that developed a technology for making building structures light-sensitive. It uses titanium dioxide, which serves as a photocatalyst similarly to the chlorophyll in plants. In the presence of oxygen and water vapors, it reacts and neutralizes pollutants in air. It converts nitrogen oxide, which is harmful to human health into harmless nitrates. As reported by Bolte [145], TioCem is a commercial cement that can be used in mortar for plastering the external walls of buildings to neutralize NOx and VOC (Volatile Organic Carbons), both of which are common ingredients in car emissions.

Essential Magazine [146] reports examples of buildings that absorb air pollution. The article describes buildings that have green vegetated walls, balconies and roofs. Bosco Verticale in Milan and Green 25 in Turin are examples of such buildings. The magazine also reports the Palazzo Italia in Milan, which is built with a photocatalytic (smart) cement that absorbs pollutants in the air and converts them into harmless salts. The structure of the building is designed in such a manner that cement-plastered surface areas are maximized. The building can neutralize air pollution from 100 diesel and 300 petrol engines. Essential Magazine [146] also reports the case of Hospital Manuel Gea Gonzalez in Mexico City that has a coating of superfine titanium oxide, which absorbs and neutralizes the air pollution emitted by cars in combination with daylight.

Yadav [147] reports the development of pollution-absorption bricks. These bricks are very porous and can filter air from outside the building structures, retain pollutants and emit cleaner air to the inside of the buildings. Air pollution filtration bricks are a low cost and low maintenance solution against air pollution that can be used in developing countries as well.

## 5. Mitigation Strategies to Reduce Health Impacts of UHI

UHI reduction strategies commonly take two forms [148]: (i) increase the solar reflectance (albedo) of roofs and pavements and (ii) increase green areas.

*5.1. Cool/green Roofs and Pavements/Walls*

The first measure, commonly termed as cool roofs/pavements, requires the provision of light-colored roofs and pavements to reflect the incoming solar radiation and to increase its emittance (i.e., the ability to release heat through long wave infrared (IR) radiation). The dual combination of increased reflectance and emittance keeps the surface cool by lowering its ability to capture heat. Sufficient commercial research and testing have been carried out with respect to the viability of cool roofs in the USA and Europe and well-established industrial standards for such commercial products have been in place since 2000 [149,150]. Due to their low albedo, pavements are a major contributor to UHI [151]. Therefore, using cool pavements provides an effective UHI mitigation measure. Cool pavements are available as two types of commercial products: (i) high albedo pavements and (ii) water retention/permeable pavements [152]. A review of the measured performance of high-albedo pavements revealed an average decrease in ambient temperature between 1 and 5 °C [153]. Pfautsch et al. [154,155] explored methods to cool roads and car parks with surface applications, shading with solar-cell sheets and vegetation.

A variation of cool roof is a green roof in which vegetative cover is provided on the roofs/walls of a building. A green roof helps in reducing UHI by absorbing sunlight with minimum radiation. In addition to reducing the surrounding temperature, green roofs are also effective in controlling storm runoff [156]. Green walls can be adopted as a UHI mitigation measure where steep roof slope or rooftop equipment prevents the installation of a green roof. Green walls are generally of two types: green façade (GF) and living wall systems (LWSs). Plants and creepers are rooted in the ground and are guided/attached to the walls in a GF while these are grown in a substrate material directly attached to the wall in an LWS [157]. As green roofs/walls are a part of the urban green infrastructure, these have the added benefit of being effective in reducing urban air pollution, as detailed in Section 4.2.

The adoption of the above measures has resulted in positive impacts on UHI reduction as well as illnesses related to heat stress in various US cities [158,159]. The numerical modelling carried out by Macintyre and Heaviside [85] found that cool roofs have the potential of reducing UHI intensity by 23% and heat-stress-related mortalities by 25% during heatwaves in a European city. They also inferred that the replacement of cool roofs for industrial and commercial buildings has more significant impacts in reducing UHI intensity compared to the residential properties. Kim et al. [160] found that the contribution of urban green roofs towards UHI mitigation in an urban center dominated by high-rise buildings is marginal compared to an increase in the tree canopy and grass coverage areas.

## 5.2. Increase in Green Areas

Urban infrastructure in all but desert cities is built at the expense of natural vegetation or water bodies. Therefore, it is intuitive that an increase in the green or blue areas in the city will help in mitigating UHI. Green areas can take the form of grassy fields, shrubs or canopy-forming trees. Provisions of green areas are strongly advocated in the paradigms of Smart Growth and New Urbanism, as detailed in Section 1.2. Grass cover works similarly to a green roof in reducing the surface's temperature. Canopy trees provide shade, which helps in lowering the ground surface temperature in addition to the cooling effect provided by evapotranspiration.

Climatic conditions were also found to have a profound impact on the effectiveness of the UHI mitigation strategies. Salata et al. [161] used numerical simulations to study the combined impact of cool roofs, urban vegetation and cool pavements on UHI and noted a decrease in the Mediterranean Outdoor Comfort Index (MOCI) of 2.5 to 3.5 along with a drop of 60% in the health risk due to heat exposure. Vargo et al. [162] summarized the possible impacts of UHI reduction policies on health vulnerabilities based on race, age and income in three large US cities. They concluded that strategies need to be designed based on the vulnerabilities of the targeted segments to be effective and the customization of the solution(s) is necessary for success.

## 5.3. Public Policy Measures for Mitigating UHI

The studies noted above prove that UHI can be controlled and its related health impacts can be mitigated despite the current trend of urbanization and a general increase in temperature due to global warming. Health professionals and sanitary engineers were at the forefront of urban planning during the early years of the industrial revolution and were instrumental in controlling health risks in the urban centers caused by communicable and infectious diseases [163]. On the contrary, the post-industrial era is confronted with public health challenges stemming from chronic diseases such as heart disease, cancer, diabetes, respiratory illnesses, obesity and cerebrovascular diseases. These ailments are more or less related to the sedentary lifestyle offered by the prevailing urban built environment [164].

The exacerbation of most of these conditions during heatwaves has also been globally recorded [165]. Corburn [166] notes a disconnect between today's urban planners and public health officials and strongly advocates an active role by the public health officials in urban planning matters that promotes healthy urban living by the design of the urban built environment. Kent and Thompson [167] explored ways for an improved integration

of public health policies and urban planning practices and urged for more cooperation between these disciplines to achieve the goal of healthy and sustainable cities. The legal framework through which today's public health officials can influence the urban planning decisions is succinctly outlined in Perdue et al. [168]. Hoverter [169] compiled a list of policy recommendations for local governments to cope with the UHI threat. Hewitt et al. [170] surveyed 26 North American cities located in various climate zones and reported the requirement of cool/green roofs for new developments in about half of them. Dare [171] reviewed 307 UHI mitigation measures adopted in 20 large North American cities. He noted that although there is an awareness among local government officials to address UHI, only a quarter of the measures were framed with a clear context to achieve a certain goal. Furthermore, the quantification of the impact of these measures on human health was found to be very weak.

## 6. Conclusions

Air pollution, built environment, UHI and human health are intricately linked with car-oriented urban developments. Human health is harmed from the by-products of reduced physical activity and air pollution from motor vehicles. The matter is made worse by the UHI effect in cities made of materials that absorb and retain heat. In recent decades, a number of new overlapping urban planning and design concepts have been developed to counter this situation. These concepts promote compact development, walkability, transport choice and attractive public open spaces. These concepts have been gaining traction and have been adopted widely.

However, the fast increase in cars across the world continues. Moreover, because many cities have already been suburbanized, car reliance for travel is likely to continue. The solution to this problem, thus, lies in retrofitting cities guided by the aforementioned planning and design concepts for walkability and in technological solutions for air pollution and UHI control. This paper has presented a broad overview of both.

Of the technological solutions, the most promising for urban air pollution reduction is the replacement of petroleum-burning cars with hybrid, electric and hydrogen cars. There is a strong momentum in that shift. Countries across the world are pushing in that direction. However, this will take a few decades. In the meantime, other technologies such as green infrastructure, stricter car emissions and fuel standards, and air pollution absorbent construction material can help in reducing air pollution and UHI and consequently improve human health by reducing the prevalence of chronic diseases.

UHI and urban air pollution are both stressors for human health. The UHI can amplify urban air pollution. The total impact of individual stresses caused by the UHI and air pollution would be higher than a simple sum total of the two. UHI causes air turbulence that increases particulate matter, NOx and ozone levels in the cities. Studies have indicated a complex but direct relationship between urban pollution, UHI and deteriorated human health. However, interactions between various drivers in this relationship is not clear. Therefore, much more in-depth research that examines the synergistic relationship between urban air pollution, UHI and climate change and their combined impact on human health is needed. There is also a need for empirical research on the effectiveness of various urban planning paradigms and air pollution and UHI mitigation measures for improving human health.

**Author Contributions:** Conceptualization, A.P.; Data curation, A.P. and M.T.C.; Investigation, M.T.C.; Resources, A.P. and M.T.C.; Writing—original draft, A.P. and M.T.C.; Writing—review & editing, M.T.C. and A.P. All authors have read and agreed to the published version of the manuscript.

**Funding:** This research received no external funding.

**Informed Consent Statement:** Not applicable.

**Conflicts of Interest:** The authors declare no conflict of interest.

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
