# Peer review of "Urban Air Pollution, Urban Heat Island and Human Health: A Review of the Literature"

_sustainability, doi:10.3390/su14159234_

Round 1

Reviewer 1 Report

The scientific novelty of the proposed study lies in analyzing and highlighting the relationship between urban heat island (UHI) and urban pollution island (UPI) from the perspective of urbanization including: urban planning, traffic.  

A scientific prerequisite for creating comfortable living conditions in the city will be to understand the interaction of urban heat islands (UHI) on fine dust dispersion.

The results of the scientific project will determine the impact of urban morphology on urban heat islands and urban pollution islands and help planners identify measures to minimize the impact of urban heat islands at the same time ensure air quality in urban areas. On this basis, analytical solutions are proposed to establish thermal balance and at the same time reduce the level of air pollution as components of comfortable living conditions.

Author Response

We found your comments very helpful. We have made significant improvements in our paper in light of your guidance including:

  1. Addition of Section 1.3 about organization of paper and analysis and breakdown of used references w.r.t. subject matter and time period.
  2. Significant editing improvements for better readability.
  3. Reorganization of Section 2 to include problems caused by effects of urbanization. It was organized into separate sections of air pollution and UHI phenomena and their impacts on human health.
  4. Previous Section 2.4 has been transformed into an independent Section 3.
  5. Previous section 3.4 has been transformed into an independent section 5 with further subsection.
  6. Conclusions section has been rewritten to identify research gaps.
  7. 38 new/additional references have been cited.

We have attached revised (track-changes) paper with this message

Reviewer 2 Report

This article carries out a systematic review of papers related with urban pollution, UHI and Human Health. The review includes references related pollution and UHI effects, the consecuences of them, relation with health and urban planning, as well as some technological solutions for improving the situation. The literature shown is up to date (most of them from this century). I consider that the review is very exhaustive, with more than 140 references among papers, reports and policy documents. 

The paper is quite well conducted, presenting an overview of the very last contributions to this research field. I recommend to accept this paper with minor revisions, including the following: Although is essential to describe the different existing references, it may be useful to have some type of summary (maybe a table, a graph) with the statistics of references for each section cites, main research lines, links among them and so on) as well as for the technological solutions (adoption, proposals, etc). It makes it possible to illustrate trends, gaps and possible relationships to complement the review. 

Also, references should be uniformed. For example, first reference introduces the words "volume" or "pages", but not in others. I miss dois, dots at the end of every reference. Also with fonts, styles (sometimes the journal name is in italic, not others....)

Author Response

(The authors gave the same response as above.)
